# Caring in the silences: why physicians and surgeons do not discuss emergency care and treatment planning with their patients — an analysis of hospital-based ethnographic case studies in England

Karin Eli [ID],[1] Claire Hawkes [ID],[1] Gavin D Perkins [ID],[1,2] Anne-Marie Slowther [ID],[1] Frances Griffiths [ID] [1]

¹Warwick Medical School, University of Warwick, Coventry, UK
²University Hospitals Birmingham NHS Foundation Trust, Birmingham, UK

**Correspondence to**
Dr Karin Eli;
Karin.Eli@warwick.ac.uk and Professor Frances Griffiths;
f.e.griffiths@warwick.ac.uk

## ABSTRACT

**Background** Despite increasing emphasis on integrating emergency care and treatment planning (ECTP) into routine medical practice, clinicians continue to delay or avoid ECTP conversations with patients. However, little is known about the clinical logics underlying barriers to ECTP conversations.

**Objective** This study aims to develop an ethnographic account of how and why clinicians defer and avoid ECTP conversations, and how they rationalise these decisions as they happen.

**Design** A multisited ethnographic study.

**Setting** Medical, orthopaedic and surgical wards in hospitals within four acute National Health Service trusts in England.

**Participants** Thirty-four doctors were formally observed and 32 formally interviewed. Following an ethnographic case study approach, six cases were selected for in-depth analysis.

**Analysis** Fieldnote data were triangulated with interview data, to develop a 'thick description' of each case. Using a conceptual framework of care, the analysis highlighted the clinical logics underlying these cases.

**Results** The deferral or avoidance of ECTP conversations was driven by concerns over caring well, with clinicians attempting to optimise both medical and bedside practice. Conducting an ECTP conversation carefully meant attending to patients' and relatives' emotions and committing sufficient time for an in-depth discussion. However, conversation plans were often disrupted by issues related to timing and time constraints, leading doctors to defer these conversations, sometimes indefinitely. Additionally, whereas surgeons and geriatricians deferred conversations because they did not have the time to offer detailed discussions, emergency and acute medicine clinicians deferred conversations because the high-turnover ward environment, combined with patients' acute conditions, meant triaging conversations to those most in need.

**Conclusion** Overcoming barriers to ECTP conversations is not simply a matter of enhancing training or hospital policies, but of promoting good conversational practices that take into account the affordances of hospital time and space, as well as clinicians' understandings of caring well.

## STRENGTHS AND LIMITATIONS OF THIS STUDY

⇒ The study's fieldwork methodology allowed for rich data collection.
⇒ Using an ethnographic case study approach, the researcher triangulated fieldnote data with interview data, developing a 'thick description' of each case, and thereby producing an in-depth analysis.
⇒ The findings are limited by the study's multisited design, which meant the researcher could not spend a substantial amount of time on one ward or in one National Health Service trust, thus limiting the temporal depth of the data collected.
⇒ While the study was designed to include medical, orthopaedic and surgical ward areas, specific wards were selected by the local principal investigator in each site, such that findings were not directly comparable across hospitals.
⇒ The researcher did not cover night shifts, and therefore, additional context for deferred and avoided emergency care and treatment planning conversations might have been missed.

## INTRODUCTION

Emergency care and treatment planning (ECTP) discussions are among medicine's most difficult conversations. Like advance care planning (ACP), ECTP processes focus on recording people's preferences for their future care, in the event they might become too unwell to express their wishes. Key to ECTP and ACP processes are conversations between clinicians and patients about future treatment options. Despite increasing emphasis on integrating ECTP and ACP into routine medical practice in the USA, the UK and Canada,[1] clinicians sometimes delay or avoid these conversations.[2 3] Reasons tend to be both structural and interpersonal. From a structural perspective, time pressures, lack of conversation skills training and uncertainty

about how treatment plans will be operationalised have been identified as barriers.[4 5] From an interpersonal perspective, clinicians' concerns about communicating prognostic uncertainty, maintaining trust and rapport, and managing patients' and relatives' expectations and emotional reactions are frequently cited as the main reasons for delaying or avoiding conversations.[2 6 7]

While studies have identified barriers to ACP, ECTP, and do not attempt cardiopulmonary resuscitation (DNACPR) conversations, little is known about the institutional logics—that is, the combination of clinical cultures and hospital structures and policies—underlying these barriers. One exception is Pavlish et al's[8] California-based study, in which they suggest that interpersonal barriers to DNACPR conversations are institutionally grounded, and that 'a culture of avoidance' drives clinicians to defer these conversations. They argue that, in emphasising time-efficient practices, rotating teams continuously, and operating through power hierarchies, clinical systems encourage practitioners to avoid ethically fraught situations, both those negotiated vis-à-vis patients (eg, DNACPR discussions) and those negotiated vis-à-vis colleagues (eg, disagreements about future care). Their analysis, therefore, suggests that clinicians' concerns about trust and rapport find an institutional framework that translates them into deferral rather than action.[8]

Although Pavlish et al's[8] study offers a substantial step toward theorising the deferral and avoidance of difficult conversations, their analysis is limited by a reliance on clinicians' reflections and rationalisations of past practices. In the last decade, several studies have used ethnographic observation methods to explore clinical decision-making about DNACPR and other treatment escalation conversations.[9–13] These studies have shown that observational research can locate clinicians' decisions about DNACPR and attendant conversations, as well as their implications for patient care, within the everyday flows of clinical practice, thereby adding an analytic dimension that cannot be captured by interview and focus group data alone.

This study's aim is to develop an ethnographic account of how and why clinicians defer and avoid ECTP conversations, and how they rationalise these decisions as they happen. Extant ethnographic studies have focused on completed forms or conversations as a main object of research; however, as we recently found,[2] forms often remain unfilled and conversations undone, at times despite doctors' stated intentions to complete them. To understand these seemingly blank spaces in clinical decision-making, we draw on hospital-based observations and interviews conducted as part of our evaluation of the Recommended Summary Plan for Emergency Care and Treatment (ReSPECT). Recently implemented in numerous National Health Service (NHS) trusts across the UK, ReSPECT is a new initiative developed in 2016 by a working group chaired by the Royal College of Nursing and the Resuscitation Council (UK). ReSPECT was created in response to shortcomings and inconsistencies in DNACPR processes, and is an ECTP process that

offers a patient-centred alternative to DNACPR.[14] The ReSPECT form is designed to prompt discussion between clinicians and patients about a range of emergency treatment options, of which CPR is only one. Given to patients and kept in their possession, the completed ReSPECT form transitions with the patient as a stable, yet modifiable, record. The affordances of the ReSPECT form thus exceed those of DNACPR forms: it allows clinicians to record patients' wishes and values alongside a list of treatments to consider in an emergency, while enabling patients to communicate their wishes across healthcare settings, including primary and community-based care, hospitals, ambulance services and nursing/care homes. As part of evaluating this new process, we were particularly interested in observing ReSPECT conversations between hospital-based clinicians and patients, as this is where most ReSPECT forms have been issued during the early implementation phase. However, as we soon learned, this intended focus on observing conversations glossed over the long stretches of time between conversations, and the deferred conversations contained therein.

We approach our analysis of clinicians' deferral and avoidance of ReSPECT conversations through a conceptual lens of care. Informed by recent sociological and anthropological theorisations of care as a mode of attentiveness cultivated and expressed through practice,[15 16] we explore how non-conversations are emplaced within clinicians' everyday practices of clinical care. Following Lavis et al[15], we acknowledge practices of care as both personal and institutional. In other words, we recognise that while care is practised by individuals (in this case, clinicians), it is also subject to broader professional and institutional framings. As such, practices of care may be driven simultaneously by a clinician's own attitudes and by the possibilities and standards articulated by medical education, best practice guidelines, and hospital structures and policies. Our analysis foregrounds this tense hybridity of care—felt and prescribed, improvised and codified—as negotiated between doctors and institutions, and as expressed through the 'non-doing' of ReSPECT conversations. Thus, rather than use a binary framework that positions ReSPECT conversations as 'careful' and non-conversations as 'neglectful', we aim to understand what the seeming silence of deferral and avoidance reveals about the logics of clinical care, both personal and institutional, that make certain ReSPECT conversations possible and precludes others.

## METHODS

As part of the larger ReSPECT Evaluation Study, we conducted a qualitative study focused on understanding when, why, and how hospital-based clinicians use the ReSPECT process in everyday clinical practice.[17] Designed as a multisited clinical ethnographic study, this study employed observations and interviews with clinicians (physicians and surgeons), patients and patients' relatives in six acute NHS trusts in which ReSPECT was

implemented. In each hospital, we aimed to conduct observations and interviews in at least five ward areas—three medical, one orthopaedic and one surgical—and thereby account for potential differences between specialities.

The first round of data collection took place in two hospitals between August and December 2017. The observations followed a shadowing framework, with the study's research fellow (a postdoctoral researcher with a PhD in public health) following consultant clinicians (senior fully qualified doctors) during ward rounds. After observing the clinicians, the research fellow conducted semistructured interviews with them, using a topic guide developed by the study team (A-MS, CH, FG and Cynthia Ochieng) as informed by the literature and by discussions with the study's clinical coinvestigators. Analysing the initial data, we recognised that ReSPECT conversations were not confined to clinical encounters during ward rounds. Additionally, based on these preliminary results, A-MS, CH, FG and KE refined the existing topic guide and developed new questions (the topic guide is available as online supplemental file). The second round of data collection, which included four sites (five hospitals in four NHS trusts) and took place from April 2019 to January 2020, followed an expanded observation framework. This expanded framework allowed KE, the study's senior research fellow and a medical anthropologist, to observe ReSPECT conversations outside ward rounds (eg, during visiting times), conduct additional observations of ward practices and have informal conversations with staff members. Through this, KE situated ReSPECT conversations within broader networks of people and practices. At each of the four sites, fieldwork took place during weekdays (4–5 days a week), usually covering a full day shift and sometimes extending into evening shifts. The length of fieldwork was determined based on ward area saturation, and ranged from 2 to 6 weeks.

The analysis is based on observations and clinician interviews conducted in the second round of data collection. We focused on these data as they reflect a more inclusive ethnographic approach compared with the data collected at the first two study sites. Across these five hospitals, 34 clinicians were formally observed and 32 were formally interviewed. The interviewed clinicians included 31 of the observed clinicians, and one consultant-level clinician who was not observed individually, but whose ward had been observed. The 32 participants included 22 men and 10 women; most (n=20) were consultant level, but the sample also included junior level (n=5) and middle-grade (n=7) doctors. Descriptive data concerning the interviewed clinicians appear in table 1.

ReSPECT conversations were at the heart of the observations and interviews. Being present at the right place and time for a ReSPECT conversation meant regularly moving between assigned wards and checking with clinicians in key moments throughout the day (eg, after the morning ward round, after the midday board round [multidisciplinary team meeting], after patient intake

**Table 1** Participant characteristics—interviewed clinicians, total sample (n=32)

| Study site | A: n=9<br>B: n=7<br>C: n=10<br>D: n=6 |
|---|---|
| Specialty | Geriatric medicine: n=7<br>General surgery*: n=5<br>Acute and emergency medicine: n=4<br>Critical care: n=4<br>Orthopaedics: n=4<br>Acute geriatric medicine: n=2<br>Acute stroke: n=2<br>General medicine/gastroenterology: n=1<br>Renal medicine: n=1<br>Respiratory medicine: n=1<br>Surgical assessment: n=1 |
| Physician/surgeon level | Junior: n=5<br>Middle grade: n=7<br>Consultant (senior specialist): n=20 |
| Gender | Women: n=10<br>Men: n=22 |

*General surgery refers to hepatobiliary surgery and colorectal surgery.

in the afternoon). To facilitate this, in two hospitals, KE was given a pager; in another hospital, a research nurse working with KE provided his own bleep number to colleagues; in all hospitals, KE provided her mobile number to clinicians on the assigned wards. However, no technology could replace waiting on the wards. Indeed, the great majority of KE's time was spent in waiting. While waiting, KE consistently took handwritten notes, exploring the mundane practices that unfolded around her, documenting staff members' informal explanations of why certain conversations would take place and others would not, and weaving together a tapestry of interactions between structures, temporalities, and practices in which ReSPECT conversations occurred—or did not occur.

This paper focuses on ReSPECT conversations that did not occur. We have previously described the range of ReSPECT conversations that occur following acute hospital admissions, and the factors that facilitate these conversations.[17] However, because periods of waiting dominated the collection of data, and because waiting for a planned conversation did not always lead to an actual conversation, focusing on ReSPECT conversations alone as the key object of research would miss the realities between conversation events. Following an ethnographic case study approach, KE selected six cases of waiting times and of ReSPECT conversations that did not actualise (see table 2). To select these case studies, KE reviewed the corpus of data and noted where deferral and avoidance of conversations were observed and discussed. KE then classified these into discussions into minor and major categories, with the latter category capturing discussions where deferral and avoidance dominated the observation and

**Table 2** The selected six cases (physicians/surgeons: n=7)

| Case | Site | Specialty | Physician/surgeon level |
|---|---|---|---|
| 1 | B | Acute and emergency medicine | Consultant |
| 2 | A | General surgery | Consultant |
| 3 | B | Acute and emergency medicine | Consultant |
| 4 | B | General surgery | Middle Grade |
| 5 | A | Geriatric medicine | Middle grade and consultant |
| 6 | D | General surgery | Consultant |

the interview that followed. She triangulated data drawn from her fieldnotes with data drawn from her formal interviews with the clinicians involved in these six cases, in which they reflected on why these conversations did not occur. This allowed us to develop a 'thick description' of each case,[18] and thereby produce an in-depth analysis of the underlying logics that thread these cases together. While data analysis was performed by KE, in keeping with ethnographic analysis methods, the findings were discussed and refined in team discussions which included all coauthors. Data analysis software was not used.

The study was reviewed and approved by a National Research Ethics Service Committee. Observations were handwritten by KE and selected relevant excerpts were typed up for this analysis; the interviews were digitally recorded and transcribed by a professional service. All participating clinicians provided written informed consent prior to being interviewed. Their team members were aware of the researcher's presence, and obtaining individual consent from them was unnecessary because they were not the focus of the observations and were not interviewed. Pseudonyms are used throughout this paper and identifying details have been omitted or altered. In keeping with ethnographic research practice, the findings are written in the first person, to acknowledge KE's experiential and interactional roles in the case studies analysed.

### Patient and public involvement

The ReSPECT Evaluation Study is supported by a patient and public involvement advisory group, which has provided feedback on the overall study design, consent models, qualitative data collection plans and initial qualitative findings from the first round of clinician interviews.

### FINDINGS
#### 'Is an acute situation the right time?'

'ReSPECT is usually done on admission' was the response I received, time and again, when asking about planned ReSPECT conversations in receiving wards. Following several days of observing, checking, and mostly waiting, I joined D1, an emergency medicine consultant, for an afternoon in Site B's emergency department's resuscitation area. Soon after, two young patients with major traumas were admitted, and the area was teeming with doctors, nurses, radiographers, porters and police officers. Amidst the ongoing team efforts at stabilising these two patients, other patients, including elderly people with various types of cardiopulmonary distress, were being brought in and taken out as soon as a bed became available elsewhere. At around 5:30 pm, there was a temporary lull in activity. D1 and I were sitting at the staff desk, and he turned to me and described the cases.

> I asked if he was planning a ReSPECT conversation, and he said no. The woman opposite me, he explained, was 'in a bad way', but her condition could easily be ameliorated through administering particular interventions, which would return her to the way she was before she came here. (Fieldnotes)

The afternoon stretched into evening, and then night. By the time I left, I had been in the resuscitation area for 9 hours, during which, despite the area being at full capacity, only one ReSPECT conversation took place. When I met D1 for an interview, I mentioned the conversation we had at the staff desk, and asked if he could expand on his reasoning for not holding ReSPECT conversations with these patients. He returned to the case of the elderly woman:

> Well, actually, in her, in her case, the pathology that had caused all of that derangement was expected to be quite reversible (…) So if, for example, she was to go into cardiac arrest, it would make perfect sense to try and resuscitate her because there's a good chance that we'd be able to.

Likewise, D1 said, the trauma patients had 'reversible pathology' and were 'not cases that we'd consider limiting treatment for or not doing CPR for'. He then mentioned additional cases of patients with drug and alcohol overdoses, where 'certainly there'd be no reason to limit treatment for them because we, because we, we expect that resuscitation, and whatever level of care we might need to provide, has a good chance of a successful outcome'.

While D1 cited reversibility of pathology as his main reason for not holding ReSPECT conversations, this was embedded in a broader contextual understanding of the ReSPECT process, its place in medical care, and the capacity to realise its potential in the emergency medical setting. After I mentioned another patient who did not have a ReSPECT conversation—a patient whom D1 identified as having reversible pathology, but also potentially reduced quality of life—D1 said:

> I find that when you're having a ReSPECT discussion there's two reasons why you might have it. One is futility of resuscitation, and there was no perception that any of these cases had futility. And the other might be that there's a quality of life issue, but I find that second one a bit harder because that's a real judgement,

and who should make that judgement? You know, really the patient is the best person to make that judgement, but is an acute situation the right time to have that when they're, they've got what's almost certainly compromised mental capacity?

As D1 explained it, when resuscitation is determined to be futile, this gives ReSPECT conversations a clarity and a raison d'être. Conversations about quality of life, on the other hand, become ethically slippery in the emergency setting. 'To me', D1 said, 'it feels inappropriate to impose that discussion on somebody if it's only about quality of life'. For D1, not having ReSPECT conversations with patients he expected would survive their hospital admission was an act of care, responding to and respecting patients' vulnerabilities within the temporal, spatial and contextual pressures of an emergency admission. When I asked him, '[w]here would you say might be a good setting to have that kind of quality of life discussion?', he replied, '[b]efore the patient gets ill':

in the cold light of day when they've got a bit of time and they're well and they've got time to consider things, we've got their values and preferences fed into this discussion about what we might do in the event that things deteriorate.

D1 added that, when a patient was admitted with a pre-existing form, he used it to inform discussions with the patient about treatment escalation. For him, patients' values and preferences were key to making decisions about treatment escalation; however, the emergency department was not the right space in which to elicit values and preferences for the first time.

### 'In the wrong place mentally to have that discussion'

I was about to leave for the surgical ward in Site A, to observe a pre-planned ReSPECT conversation, when I received an email from the consultant surgeon, S1, saying the conversation was not to be. Later that day, S1 and I met for an interview. I opened the interview by asking S1 how she decided to have the conversation with this patient, and why the conversation did not happen. S1 began by describing the patient's condition. She had a terminal illness, S1 said, '[s]o we were gonna have the discussions about planning further treatments, further options and what's agreeable and acceptable to her'. 'And she knew that she was going to have the discussion today?' I asked. 'She knew that she was going to have the discussion today', S1 said,

but, unfortunately, she's been in pain all night and had a bit of disruption on the ward and she's a bit tearful. (…) she just seemed a little bit in the wrong place mentally to have that discussion. So I spoke to her and said that I would have it with her maybe tomorrow, with her husband, and, and she said she would like to have 24 hours of thinking and, and have it tomorrow.

I asked if the patient knew where her condition was heading, and S1 said the patient knew she was dying. Knowledge, however, did not equal readiness, and S1 deferred the conversation as a responsive act of care.

This deferred conversation hinted at the wider phenomenon of non-conversations on surgical wards. ReSPECT conversations rarely happened on surgical wards; the loss of this conversation, after weeks of waiting to observe one, meant that I completed my fieldwork at Site A without observing any ReSPECT conversations on the surgical ward. When I asked S1 if surgeons were particularly reluctant to have these conversations, she laughed and said:

Yeah. Definitely. Definitely. Because our patients very rarely die. (…) they'll come in with something we can treat and we send them home. So I, I think surgeons specifically don't deal with death very well, we see that as failure, we see that as failure of us, particularly in someone who dies in the post-operative period, we find that very difficult.

'And so when you do have a difficult conversation', I asked, 'how does that make you feel?'

Well, it's one of those situations that you, you, you do, you dread it, you absolutely dread it before you're going in, you, because it doesn't come naturally, you, you, you can occasionally stumble over your words and make it sound less fluid than you want. You can say the wrong thing, use the wrong terminology, which is us falling back to our medical knowledge. So it, it, I think there's a lot of anxiety going into that conversation and a lot of anxiety afterwards.

With the death of a patient deemed a professional (and personal) failure, and with ECTP conversations perceived as an acknowledgement of impending death, S1, like other surgeons I interviewed, described these conversations as requiring a skillset surgeons did not have. As described by S1, ECTP conversations challenged her professional competence, and she found herself on an unsure interactional footing when initiating these conversations. The dread she described, however, related not only to stepping outside a professional comfort zone, but also—and centrally so—to fear of upsetting patients, and thereby caring for them wrongly or inadequately.

### 'If they're not going to die imminently, I normally give them time'

The acute medical unit (AMU), where, according to clinical staff, most ReSPECT conversations occurred, was a cornerstone of my observations in Sites B and C. Yet, despite spending considerable time on these wards, I only observed two conversations. In part, this could be attributed to coincidence—to consultants forgetting to call me into a conversation, to no new patients being admitted over the course of several days, to my being on another ward just as a ReSPECT conversation was taking place, to an on-call doctor assiduously conducting conversations during twilight hours (9 pm to 9 am). But

coincidence only went partway in explaining my sparse observations. In Site B, I spent the majority of my observation time at the AMU, including three consecutive days of shadowing lengthy post-take ward rounds where no ReSPECT conversations took place. It was on this ward that I experienced most acutely how everyday infrastructural contexts interweave with decisions about ReSPECT conversations.

Joining a sizeable ward round entourage of six or seven junior doctors, medical students and a pharmacist, I followed the consultant, D2, as he made his way from the hot desk to the patients' bedsides and back again. With three chairs and three desktop monitors, the hot desk was nearly always overcrowded in the mornings, with staff members playing a seemingly endless game of musical chairs as they attempted to view patients' records on the monitor while juggling two ring binders for each patient (one with medical notes, the other with pharmacological notes). Like most of D2's entourage, I spent most of the ward round standing in the misshapen semicircle that formed around him as he presented each case. With the corridor thus congested, I frequently found myself moving aside in order to allow staff members to walk through. This embodied experience of resource limitation was magnified when I followed D2 to the first board round of the day, where I wrote:

> We're entering the MDT room which is standing room only. The day room is not available (there's an interview there). People are moving chairs and tables to be able to close the door. (Fieldnotes)

D2 and another colleague mentioned ReSPECT when presenting four patients, at least three of whom were classified as end-of-life. Of these patients, three had a form in place, and D2 said a palliative team should be called in for the one who did not. ReSPECT, then, was highlighted as a key form designated for, and identified with, the terminally ill. Having attended this board round, I was surprised when, the next day, at a board round led by another team, ReSPECT received no mention, despite some patients having multiple comorbidities. 'The main focus', I wrote in my notes, 'was on deciding which patients could be discharged, when, and which would need a bed on a receiving ward'. Bed capacity was a main concern in the board round that followed, too. Led by D2, the board round was interrupted when a nurse entered to ask which patient could be moved to another ward, so they could move another patient into AMU, urgently. The next day's post-take ward round was, in D2's words, 'chaotic', and his long list included new patients alongside several others who were very unwell. No ReSPECT conversations took place during this ward round, and no prospective conversations were mentioned during the board round that followed, though patients with organ failure were reviewed.

When I interviewed D2, I asked why no ReSPECT conversations took place. 'In our last three days', he said, 'luckily there was no sick patients who needed a ReSPECT

form or the people who were sick already had one in place (…) so I didn't have the chance to discuss'. D2 defined 'sick patients' as those 'who were going to imminently die by the looks of things'. His equating of ReSPECT with an immediately terminal prognosis was linked to the pace of clinical practice at the AMU. When I asked him if high patient turnover impacted on ReSPECT conversations, he spoke about the extended time for discussion on the receiving ward where he also worked, and then said:

> I agree, in acute there is less of an opportunity to discuss because the turnover is so huge and patients move so rapidly to different wards. (…) But if, like, we're in a situation when we think patient is acutely unwell, he is probably not going to make it, he is likely going to pass away in next couple of hours, so then of course we do discussions with patient families.

Initiating a ReSPECT conversation at the AMU involved a triaging process, in which patients became candidates for ReSPECT if they could not be stabilised and moved to a receiving ward. The logic behind limiting ReSPECT conversations to those in immediate need was captured in D2's description of two types of patients with whom ReSPECT 'needs to be discussed'. In the first group, were 'patients (who) already know that they've got an irreversible, advanced, incurable disease'. Discussing ReSPECT with patients from this group, D2 said, was 'easy'. The second group, however, presented additional challenges:

> Now, doing ReSPECT form in patients who have been suddenly been diagnosed [with an incurable] illness is challenging (…) because till yesterday they were fine and now we're telling them you've got an incurable illness. So sometimes my approach in these situations, if they're not going to die imminently, I normally give them time. (…) Maybe once, two, three sittings and often involve multiple teams.

Easing patients into a ReSPECT discussion took time and involved a series of conversations with multiple staff members. However, this stretched the scarce temporal, spatial and staff resources on the ward. Unable to give more AMU patients the time and careful discussions they needed to come to terms with their prognosis and treatment plans, D2 therefore reserved ReSPECT conversations for those at the end of life.

### 'The terminally ill patient will unfortunately go down in the priority'

Following a board round on a surgical ward in site B, I was sent to the critical care ward area along with a middle-grade surgeon, S2, and a junior doctor. When S2 was paged and asked to see a patient at the high-dependency unit (HDU), the consultant said I should join him, as this was a more promising location for a ReSPECT conversation. Walking and talking on the way to HDU, S2 said that,

> during the day, people are running around like headless chickens, trying to fix people. So if a patient is

'unsalvageable', they leave that conversation for the on call team after hours. (Fieldnotes)

At HDU, I waited while S2 and the junior doctor examined a patient. When they emerged from behind the curtain, S2 said he didn't bring me into the conversation because 'they were discussing chemotherapy with the patient, so he's a few steps away from a ReSPECT conversation'. From the HDU, we walked to the intensive therapy unit (ITU), where the junior doctor told me the first patient was 'just a normal patient'. I waited, standing in the corridor and writing notes. Then, in what turned out to be a critical moment in my observations, S2 took me to see a patient at the end of the corridor. He told me this patient was dying of metastatic cancer and in need of a ReSPECT conversation, but that this would be a discussion with the patient's relatives, as the patient was ventilated.

> I asked if her family was around, and he said let's see. We walked toward her bed area and the curtain was drawn. I stopped before reaching the bed area, and [S2] approached and moved the curtain lightly, with one finger. He looked inside, let go of the curtain, and said, 'too late'. I asked [if] she's still there, and he said she's just died, and she's covered. As we walked back, he told me it's a sad way for a person with metastatic cancer to die. He said he was called by an ITU doc yesterday (…) and knew he needed to have a ReSPECT conversation for this patient, but then other patients' treatment needs got in the way, and the conversation was postponed and then didn't happen, eventually. (Fieldnotes)

Two days later, when I interviewed him, S2 referred several times to 'the window for ReSPECT discussion'. With this patient, he said, this window did not exist. He explained that, as a surgical patient, she had been admitted for a reversible condition. So, although she also had metastatic cancer, it was 'not appropriate for people [surgeons] to have that conversation with her'. As her condition deteriorated and she was admitted to ITU, she lost consciousness and this sealed any possibility for discussion. However, the lack of a discussion 'window' was not the only reason this patient died without having had a ReSPECT discussion. Explaining why ReSPECT conversations were infrequent on the surgical ward, S2 said:

> …in a surgical speciality, for instance, if you've got 60 patients on the ward, life-threatening issues, which are recoverable or treatable X number, and the terminally ill patient unfortunately will go down in the priority of that clinician's time allocation that day.

I reminded S2 that, when we went to see the patient, he told me he wanted to have a ReSPECT discussion for the patient earlier, but ran out of time. He confirmed this, saying that

…in that time I was here till 8 o'clock with other things, so it kept being prioritised lower down on the priority [list].

A second, key reason for deprioritising ReSPECT conversations concerned clinical skills. To become a consultant surgeon, S2 said, you need many years of training. However, surgeons were not equally equipped to have difficult discussions with people whose lives they could not save:

> even though their [patients'] emotional need, psychological need, all sorts might be very high, desperately suffering person, but you put them in a situation where the group of people delivering [the] service are not trained to deal with it, not equipped to deal with it, so that goes down in the priority.

S2 prioritised caring for patients who could engage his training—patients for whom he could care professionally. Prioritising 'salvageable' patients at the expense of terminal patients, therefore, was not merely a calculated allocation of resources to those most likely to survive, but a reflection of S2's concerns over providing the care he had been trained to give.

This logic also informed S2's decision not to have a ReSPECT conversation with the first patient he saw at the HDU during my morning of observations—the patient with whom he discussed chemotherapy. After describing the patient's terminal diagnosis and short prognosis, S2 said:

> So, by definition, if you go with the lectures about palliative care, ReSPECT and all that, that's the time to have the discussion. But go back to the same thing. He was well one week ago, he was working. Now he has got [terminal] cancer. So I didn't feel that it right for me to talk to, is right for me to then talk to him about how do we end this, how do we plan for expected decline in his function.

Notably, in his response, S2 juxtaposed 'lectures about palliative care' with the reality he confronted as a surgeon. He implied that recommendations about timing ReSPECT conversations were doubly detached from his clinical practice, reflecting classroom-based rather than practical knowledge, and grounded in a specialty other than his own. Facing this patient, S2 felt that palliative care standards clashed with his own ethics of care, which he described as informed not only by the patient's temporal distance from the death toward which he was heading, but also by his temporal distance from the regular life he left behind. 'He will be shell-shocked if I told him that, 'So you will, you know, be dead in six months', S2 said. He added that, on numerous occasions, he had overheard patients making plans they would not live to fulfil:

> No, I didn't tell them not to plan those things, because I felt, I felt whatever false hope that they are

carrying is probably better than dash it all. Is it because I don't have the time, I don't have the skill? I don't know.

In this apparently ambivalent explanation, S2 wove together concerns over patients' feelings, experiences of time pressure, and insecurities over lacking the necessary conversation skills. Running through these was a central concern with caring adequately—at the right moment, with enough time, and with the right skills. Without sufficient time and skills, the right moments for ReSPECT conversations eluded S2, such that non-conversations became acts of care.

### 'Unless it's clinically urgent, I don't want to make it an overly dramatic thing'

Early in my observations at the geriatric ward in Site A, I experienced a day of waiting that ended with three deferred ReSPECT conversations. The afternoon began with the doctor in charge, D3, telling me that a conversation with a patient's family was expected. Soon thereafter, D3 explained the conversation would be postponed because the patient's relatives expressed 'disagreements with each other and with the clinical team'. This conversation was to be the third in as many days. As I wrote in my notes, D3 explained that 'the conversation often ends up taking three or four days because they want to include all family members, but relatives tend to disagree and have unrealistic expectations'.

More than two hours after this attempted conversation, D3 called me into her office to observe a ReSPECT phone call with a patient's son who visited 2.5 hours earlier, but whom she could not meet because she was caring for other patients. She gave me the single chair in her small office, sat on the desk, and made the call, which went to voicemail. D3 then said she will wake up a patient with whom she wanted to start a ReSPECT conversation. However, on exiting the office, D3 was confronted by another patient's relative. In rather urgent tones, the relative asked D3 to have a look at the patient. She agreed, turned to me, and said she'll be back. About half an hour later,

[D3] came back and told me I should probably leave now as well. Her shift was over and her time was taken up by tending to [the patient]. She wasn't going to start a ReSPECT conversation now—she'd missed her window. (Fieldnotes)

As the only doctor consistently on the geriatric ward, D3 was frequently called in multiple directions at once. She was not in the habit of deferring ReSPECT conversations—on other days, she called me to observe conversations she held with patients' relatives. However, weighing the immediacies of patient care against the sometimes-multiple, uncertain temporalities of ReSPECT conversations, she prioritised treatment delivery.

D4, a geriatrics consultant, was another clinician I observed on the geriatric wards in Site A. Over the course of 3 weeks, I repeatedly checked with him—before, during, and after ward rounds—to see if a ReSPECT conversation was expected. After my final attempt at observing a conversation, D4 let me know that, again, no ReSPECT conversation was on the horizon. At the interview, D4 offered several reasons:

Firstly, some of the patients, those conversations had, had already happened earlier in the admission. (…) And if it's happened once then duplicating that sorta conversation, which can be a difficult conversation, is clearly unnecessary, potentially distressing. (…) There'll be some patients where that conversation wasn't necessary (…) And there'll be some patients where the conversation was a good thing to have, hadn't happened yet. But because of the timing of the ward rounds it wasn't the, the right time to have that conversation.

D4's response centred on emotional management and timing. Identifying 'the right time' for a conversation hinged on multiple factors. Key to appropriate timing, D4 said, was aligning ReSPECT conversations with family visits. Ensuring the involvement of family members, however, required a careful balance of timing and emotional management:

unless it's clinically urgent, I don't want to make it an overly dramatic thing, 'cause that then attaches more weight to it. So I prefer to be opportunistic. So, if we know family are coming, or are likely to be coming, I'd rather wait for them to come.

Whereas other clinicians staged ReSPECT discussions to convey their seriousness—through scheduling a conversation in a private room with a patient's relatives—D4's approach was to disinvest the conversation of its impact, smoothing it into the patient's wider care. This meant that, sometimes, ReSPECT conversations did not happen on time, or did not happen at all. Ideally, D4 explained, he would have a ReSPECT conversation with nearly every patient, but 'society isn't quite ready for this to be a normal, a normal conversation to happen at all times'. D4 acknowledged the gap between his ideal practice and current approach to ReSPECT conversations. However, in deferring or not holding ReSPECT conversations to avoid distressing patients and their relatives, he aimed to offer a careful response to patients' and relatives' perceived lack of readiness.

### 'We are usually out of time'

When I began my observations at the surgical ward in Site D, the consultant in charge told me that ReSPECT conversations usually took place later in the afternoon. Unlike the brief, routine, CPR-focused conversations that frequently took place on acute medicine wards in Site D, ReSPECT conversations on the surgical ward were complex, and involved planning and scheduling with patients and their relatives. I therefore came back to check every day for anticipated conversations. In the first week, I observed one ReSPECT conversation. In the

second week, on my third day of repeated checking with the ward, the consultant surgeon, S3, walked past me in the corridor and said there might be a discussion later that day, 'around 3'. I arrived just before 2.30pm but S3 said they would not be having the conversation after all. I understood his reluctance. For an entire week, this conversation had been pushed from one day to the next. It was to be a follow-up ReSPECT conversation with a patient who asked for more time. However, days passed and the patient did not want to revisit the discussion. Realising this follow-up conversation was not to be, I scheduled an interview with S3, to discuss why this conversation did not take place.

I returned to the ward at 4:45 pm, having scheduled the interview for 5 pm. S3 asked me to wait 20 minutes. I stood at the nurses' desk and observed dinner being assembled on trays while S3 gave instructions over the phone about a patient's continued treatment. Following this call, he turned to me and apologised, saying he had just been called to another level. He returned about 40 minutes later, apologised again, and said he needed another 10 minutes. I stayed at the nurses' desk, where two junior doctors and a surgeon were ordering a CT scan, while S3 consulted with his colleagues about a patient in the adjacent, open-doored, doctors' office. S3 then walked out and invited me to join him at the doctors' office. He sat down to type patient notes and talked to me throughout, counting down the number of patients for whom he still had to write notes. His note-writing was frequently interrupted by questions and consultations with other colleagues—some of which led to further documentation.

After about an hour of note-taking, S3 asked me to join him as he went to check on a patient on another ward. There, we met a junior doctor from the surgical ward, and, after S3 checked on the patient, we all walked back together. It was after 6:30pm.

> I asked if it's a typical day. [S3] said they try to finish by 6 but the [junior doctor] said it was typical. He said there's a never ending mountain of things and that there's always something to fill the time. I asked him if the incessant need to write notes on [the hospital's digital system] contributes to that, and he said definitely, that it's a defensive exercise, mainly to avoid litigation, because if you don't write it, it didn't happen. To clarify, I asked him if notes are more about litigation than about informing colleagues, or a mix, and he said it's a mix (but mainly about litigation). (…) I asked if ReSPECT is just another form and he said he doesn't do ReSPECT, his seniors do, but that when something like this comes up on [the hospital's digital system] it's more like 'another thing to do' rather than an important thing I have to take care of. (Fieldnotes)

It was nearly 7 pm when my interview with S3 began. Wary of being called in for another task, he suggested we have the interview at the downstairs café, though he had his phone at the ready for calls from the ward. This

was a 12-hour workday, he told me, '[b]ut, you know, we have our duties, our duty is to patients, we want to do everything… and ensure that all jobs are done before going home'. I asked S3 why no ReSPECT conversations took place in the last three days. He began by speaking about new admissions in general:

> So when we are admitting the new patients, well, we, we don't ask them about if they want to be resuscitate[d] or not. Well, we, on the basis of our clinical findings, I mean their medical history, reason for admissions, we are mostly assuming that the patient who is not, who is relatively young, has no cancer, active cancer, or who has an active cancer, but was admitted for the operation… wants to be fully treated, so receive active treatment for CPR also.

ReSPECT, S3 explained, comes into play if 'it turns out that this cancer is inoperable'. In those cases, where patients are transitioned to 'palliative measures', the goal of the discussion is to establish a DNACPR recommendation. However, he clarified,

> just before that we have an active discussion with the patient and his family about how it looks like, how it corresponds with his treatment, what are the, what are the outcomes of our treatment, of diagnosis, so it's not a, a, a short talk, it's usually a long talk in which we are discussing what was done, what needs to be done, what can be done and how it affect[s] the patient life, quality of life and life expectancy. (…) it's not an easy talk.

S3 said patients' relatives often worry that a DNACPR recommendation would lead to broader cessation of treatment, and that much of the ReSPECT discussion is about establishing 'boundaries' of treatment and clarifying what will and will not be done. This process of clarification, he said, 'takes time':

> So that's why we don't instigate this, this, this type of discussion to every patient 'cause, as you notice, we are usually out of time.

I mentioned the patient whose conversation was deferred, and asked why he decided not to have this discussion. '[W]e knew that these kind[s] of discussion are very, very delicate ones, but we need to have also the proofs for that', he said. Although the patient was visibly deteriorating, and although the patient's condition had been discussed as terminal in the initial ReSPECT conversation, S3 decided to wait for additional laboratory results that would confirm the patient's prognosis. With test results used to remove doubts about the boundaries of treatment, S3 aimed to remove a layer of uncertainty from this already-fraught ReSPECT conversation.

Evidence and overwork, therefore, were in dialogue. S3 described ReSPECT conversations as invariably lengthy, detailed and complex. His focus on laboratory evidence, as providing much-needed scaffolding to ReSPECT conversations, conveyed the weight he placed on getting each

conversation 'right'. His lack of time, however, meant that most patients did not have these conversations. Thus, as S3 described it, the deferral or avoidance of conversations that could not provide detail, clarity and optimised results was a careful choice.

## DISCUSSION: PUTTING CARE IN CONTEXT

Centring on ReSPECT conversations that did not happen, the ethnographic case studies have revealed multiple strands of reasoning that underlie the seeming silence of deferral and avoidance. For the clinicians who took part in this study, concepts of caring through silence—through not speaking to patients and their relatives about ECTP—emerged in the interlocking of temporal, emotional and structural resources. These deployments of silence, notably, diverged from notions of 'silence as an element of care',[19] which posit that intentional silence in interactions between patients and clinicians can communicate compassion, understanding and respect for lived experience. The silences in which the participating clinicians engaged were intentional, but not interactional. As acts of care, clinicians' decisions to defer or not hold ReSPECT conversations were inherently relational, 'joining up' patients and clinicians[20]; however, these decisions were unknown to the patients or the relatives concerned, and could be conceptualised as 'backstage caring'.[16] Nonetheless, as the participating clinicians described them, non-conversations were part of a larger complex of clinical practice, driven by concerns over caring well, with clinicians attempting to optimise both medical and bedside practice.

For the participating clinicians, concerns over caring well interwove with an appreciation of ReSPECT conversations as substantial and consequential. Conducting a ReSPECT conversation carefully meant attending to patients' and relatives' emotions and committing sufficient time for an in-depth discussion, while also taking contextual factors—such as the affordances of the ward—into account. As such, plans for caring well through a ReSPECT conversation were often disrupted by issues relating to time. Buse *et al* capture the clash between care practices and institutional temporalities, writing that, while care requires flexible temporal boundaries, responsive to individual needs, this flexibility becomes fraught when confronted with institutional time, which calls for the maximisation of efficiency under pressure.[16] Indeed, tensions relating to time resources and the temporality of care ran through all six ethnographic case studies. In some cases, clinicians deferred ReSPECT conversations due to lack of time for other types of patient care. In other cases, however, it was (perceived) patient time, rather than clinical time, that led to conversation deferral, with clinicians explaining they felt patients and relatives were not yet ready for the conversation. Clinicians also spoke about temporal 'windows' for ReSPECT conversations—opportune but transient times for a conversation (eg, when a patient's family visited) that could not always be seized.

Decisions not to hold ReSPECT conversations were situated not only temporally, but also spatially. As Pink has argued, care is emplaced, that is, situated within a set of local processes, materialities and practices that give it shape and meaning.[21] In the case studies, ReSPECT (non-)conversations were diversely emplaced: whereas the surgeons and geriatricians deferred or avoided ReSPECT conversations because they lacked the time for detailed and in-depth discussions, the emergency and acute medicine clinicians deferred or avoided ReSPECT conversations because the high-turnover ward environment, combined with patients' acute conditions, meant triaging conversations to those most in need.

Notably, the participating doctors' reflections centred on deferred conversations that concerned gravely ill patients. Conversations that could have been held with patients who were generally well were left largely unacknowledged. This captures a chasm between the intended use of ReSPECT and its implementation in practice. Whereas ReSPECT was designed as a universal process that confirms treatments to be provided, and not just treatments to be withheld,[14] the case studies showed that doctors mainly considered ReSPECT when patients were at imminent risk of dying. The nexus of clinical time and care that emerged in the case studies, however, provides a possible explanation for this chasm, suggesting that doctors' limited temporal resources might make it difficult to implement the inclusive vision of ReSPECT, beyond a focus on DNACPR and comfort care recommendations.

Our findings suggest that the future implementation of ECTP initiatives would benefit from insight into the clinical logics that inform ECTP conversations and non-conversations. Like earlier studies that identified barriers to DNACPR, ACP and ECTP conversations, our study found that time pressures, concerns over patients' and relatives' emotional reactions, and insecurity about communication skills contributed to doctors' deferral or avoidance of conversations.[2 4–7] However, in using an analytic lens of care, our study offers a new understanding of deferral and avoidance not as 'deficient' or 'neglectful', but as practices embedded into, and informed by, everyday clinical care. Seen through this lens, barriers to ECTP conversations are not hurdles to be overcome, but rather concerns that make sense within the structures, values and constraints that underlie clinical practice.

While the study's ethnographic methodology allowed for rich data collection and triangulation, the study's design also had several limitations. A multisited ethnographic approach was selected to evaluate the ReSPECT process in diverse ward areas across NHS trusts. However, due to this approach, the researcher could not spend a substantial amount of time on one ward or in one trust, thus limiting the temporal depth of the data collected. Had the researcher spent an extended period on one ward, richer insights about the day-to-day unfolding of ReSPECT conversations within broader clinical practices might have been developed. Additionally, while the

study was designed to include medical, orthopaedic, and surgical ward areas, specific wards were selected by the local principal investigator in each site. This meant that observations varied across the hospitals, with some PIs selecting acute wards, such as the emergency department, and others selecting receiving wards, such as geriatrics. This led to diverse findings within the sample as a whole, but due to this diversity, findings were not directly comparable across hospitals. Finally, it is important to acknowledge that the researcher did not cover night shifts, and therefore, additional context for deferred and avoided ReSPECT conversations might have been missed.

## CONCLUSION

Drawing on a multisited ethnographic study conducted in England as part of the larger ReSPECT Evaluation Study, we analysed why hospital-based physicians and surgeons deferred or avoided ECTP conversations. We found that deferral or avoidance of ECTP conversations was driven by concerns over caring well for patients and their relatives. Importantly, the findings highlight the nexus of clinical time and care. The participating doctors conceptualised a careful ECTP conversation as one that required attending to patients' and relatives' emotions and committing sufficient time for an in-depth discussion. However, because conversation plans were regularly disrupted by timing and time constraints, doctors often deferred these conversations, sometimes indefinitely. Additionally, issues related to time manifested differently across ward areas: whereas the surgeons and geriatricians deferred conversations because they lacked the time for detailed discussions, or because they missed 'windows of opportunity' for discussion, the emergency and acute medicine clinicians deferred conversations because the high-turnover ward environment, combined with patients' acute conditions, meant triaging conversations to those most in need. The findings highlight the importance of considering clinicians' logics of care in the future development of ECTP initiatives. Based on these findings, we suggest that overcoming barriers to ECTP conversations is not simply a matter of providing individual training or changing hospital policies to encourage or mandate these conversations. Rather, short of infrastructural change that enhances human, temporal and spatial resources, future ECTP initiatives would gain from promoting good conversational practices compatible with clinicians' logics of care, and with the institutional affordances where these logics are put to work.

**Acknowledgements** This research is supported by the National Institute for Health Research (NIHR) Applied Research Collaboration (ARC) West Midlands. The authors acknowledge the support of the National Institute for Health Research Clinical Research Network (NIHR CRN). We would like to thank Cynthia Ochieng for her invaluable work on the first round of fieldwork in the ReSPECT Evaluation Study. We would also like to thank the study site principal investigators for facilitating access to the hospitals, and to the participants who generously shared their time, experiences and expertise with us.

**Contributors** KE collected the data, conducted the data analysis and drafted the manuscript. A-MS, FG and CH designed the study and critically commented on the findings and on the manuscript drafts. GDP contributed to study design, assisted with accessing the field and critically commented on the findings and on the manuscript drafts. All authors reviewed and approved the final version of the manuscript. FG is acting as guarantor.

**Funding** This article presents independent research funded by the National Institute for Health Research (NIHR) under the Health Services and Delivery Research programme (project number 15/15/09).

**Disclaimer** The views expressed in this publication are those of the authors and not necessarily those of the NIHR or the Department of Health and Social Care.

**Competing interests** GDP and CH are members of the ReSPECT national working group. A-MS, FG, CH and GDP received grants from the UK National Institute for Health Research during the study.

**Patient consent for publication** Not required.

**Ethics approval** The study received ethics approval from the NRES Committee, West Midlands – Coventry and Warwickshire (REC reference: 17/WM/0134).

**Provenance and peer review** Not commissioned; externally peer reviewed.

**Data availability statement** All data relevant to the study are included in the article or uploaded as online supplemental information. Although the qualitative data in this study have been pseudonimised, it is possible that with access to raw data individuals might be identifiable. The data are not suitable for sharing beyond what is contained within the manuscript. Further information can be obtained from the corresponding author.

**ORCID iDs**
Karin Eli http://orcid.org/0000-0001-9132-8404
Claire Hawkes http://orcid.org/0000-0001-8236-3558
Gavin D Perkins http://orcid.org/0000-0003-3027-7548
Anne-Marie Slowther http://orcid.org/0000-0002-3338-8457
Frances Griffiths http://orcid.org/0000-0002-4173-1438

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
