## [Reviewer comments · BMJ Open]

ARTICLE DETAILS

TITLE (PROVISIONAL)	Caring in the silences: Why physicians and surgeons do not discuss emergency care and treatment planning with their patients – an analysis of hospital-based ethnographic case studies in England
AUTHORS	Eli, Karin; Hawkes, Claire; Perkins, Gavin; Slowther, Anne-Marie; Griffiths, Frances

VERSION 1 – REVIEW

REVIEWER	Kathleen O'Neill Yale-New Haven Hospital
REVIEW RETURNED	03-Dec-2020

GENERAL COMMENTS	Overall, the authors are to be commended for an excellent study investigating the contextual factors that contribute to the delay or avoidance of emergency care and treatment planning (ECTP) discussions in clinical practice. This is a multi-sited ethnographic study investigating why clinicians defer or avoid ECTP conversations set in medical, orthopaedic and surgical wards in five different hospitals. The authors describe the complexities of conducting ECTP conversations in great detail through six individual case studies of instances when ECTP conversations were delayed or avoided. They found that physicians described multiple reasons for delaying these conversations that encompassed time constraints, lack of training, and concern for patients' and relatives' emotional reactions to these discussions. In this, the authors make an important conclusion that tools and training alone are unlikely to improve the quality or frequency of these discussions without structural changes in the clinical environment that provide the temporal and emotional space for ECTP conversations. My main concern with this paper is some lack of clarity around a few elements that appear to be missing/incompletely explained in the methods section of the manuscript: 1. The authors mention that 32 participants were interviewed. There is not enough information on the make up of these clinicians (specifically, the breakdown by specialty, age, distribution across sites, years in practice etc). There is also little information on the structure of these interviews—who conducted the interviews? Was there a structured interview guide? If so, how was this guide developed and by whom? A table with the first set of information and figure with the structured interview guide would be a good addition to the paper. Alternatively, if additional results of this study have been published somewhere else and this information is available there, simply citing that paper would suffice.2. While a fair amount of space was spent in the methods on these interviews, it is unclear how they were used in the final analysis of
--

	this paper. The methods section states “This paper focuses on ReSPECT conversations that did not occur” and then goes on to state that “She triangulated data drawn from her fieldnotes with data drawn from her formal interviews with the clinicians involved in these cases, in which they reflected on why these conversations did not occur.” Does that mean that only clinician interviews that were involved with the cases were included (as is suggested by the methods section)? Or were all clinician interviews included in the triangulation, picking out those relevant quotes from all interviews 3. More information about the six cases described is needed: Were they all from the same site? From the same specialty/discipline? What level of training were the clinicians involved? Much of this information can be gleaned from the narrative descriptions in the results section but it is inconsistently reported. A table summarizing this information would answer many of these questions and allow for a better overall understanding of the context of the six case studies. Alternatively, a short summary paragraph at the beginning of the results section may also suffice for relaying this information. 4. Besides the statement “She triangulated data drawn from her fieldnotes with data drawn from her formal interviews with the clinicians involved in these cases, in which they reflected on why these conversations did not occur,” there was little information on the analysis of the data. Were these cases discussed at meetings (such as at a coding meeting)? If so, please describe the make-up of this group. Was there any software used to assist in the coding of data (ie: Atlas, Dedoose)? Were there any other techniques to enhance trustworthiness and credibility of data analysis such as member checking? Or was the coding, triangulation and ultimate development of the “thick description” done purely by one person (which is what the methods section suggests)? Thank you for the opportunity to complete this review and I look forward to seeing this article in print.
--	---

REVIEWER	Ting Hway Wong Singapore General Hospital
REVIEW RETURNED	18-Dec-2020

GENERAL COMMENTS	Thank you for this beautifully conducted study (both the submitted draft and the larger overall study). As an emergency surgeon with a specific interest in end of life issues, this is an important issue to address. My main concern with the study as it stands is that it only shows the barriers to ECTP (what some of us surgeons would call a "negative demonstration"), which is meaningless without counterbalance with examples of when they were successful. If, for example, another surgeon had used exactly the same reasons to push for ECTP (eg "patient is in a bad way and I worry we don't have much time to have this conversation together" / "we are expecting to be called to another major emergency and hope we could catch some time for this conversation before our schedule is full"), then the ethnographic findings would only paint half the picture. I understand word count may have been a problem, but a workaround should be found (e.g. part 1 Why they do not discuss ECTP and part 2 Why they do).
--

VERSION 1 – AUTHOR RESPONSE

Reviewer: 1
Dr. Kathleen O'Neill, Yale-New Haven Hospital

Comment 1	Overall, the authors are to be commended for an excellent study investigating the contextual factors that contribute to the delay or avoidance of emergency care and treatment planning (ECTP) discussions in clinical practice. This is a multi-sited ethnographic study investigating why clinicians defer or avoid ECTP conversations set in medical, orthopaedic and surgical wards in five different hospitals. The authors describe the complexities of conducting ECTP conversations in great detail through six individual case studies of instances when ECTP conversations were delayed or avoided. They found that physicians described multiple reasons for delaying these conversations that encompassed time constraints, lack of training, and concern for patients' and relatives' emotional reactions to these discussions. In this, the authors make an important conclusion that tools and training alone are unlikely to improve the quality or frequency of these discussions without structural changes in the clinical environment that provide the temporal and emotional space for ECTP conversations.
Response 1	Thank you for reviewing our manuscript so positively and highlighting its strengths.
Comment 2	My main concern with this paper is some lack of clarity around a few elements that appear to be missing/incompletely explained in the methods section of the manuscript:
Response 2	We have revised the methods section in accordance with your suggestions, as detailed in the responses below.
Comment 3	The authors mention that 32 participants were interviewed. There is not enough information on the make up of these clinicians (specifically, the breakdown by specialty, age, distribution across sites, years in practice etc). There is also little information on the structure of these interviews—who conducted the interviews? Was there a structured interview guide? If so, how was this guide developed and by whom? A table with the first set of information and figure with the structured interview guide would be a good addition to the paper. Alternatively, if additional results of this study have been published somewhere else and this information is available there, simply citing that paper would suffice.
Response 3	Thank you for highlighting these important omissions. Unfortunately, we did not collect data on the clinicians' ages and years in practice; however, the latter can be inferred from the clinicians' levels, which we indicate in the manuscript – junior, middle-grade, and consultant (p. 5). We have now added a table that details the participant characteristics (site, specialty, level, and gender) (Table 1). A semi-structured interview guide was used in the study. We now explain in the manuscript that this interview guide was developed by the study team, based on preliminary results from the first round of fieldwork (p. 4): “The first round of data collection took place in two hospitals between August and December 2017. The observations followed a shadowing framework, with the study's research fellow (a postdoctoral researcher with a PhD in public health) following consultant clinicians (senior fully-qualified doctors) during ward rounds. After observing the clinicians, the research fellow conducted semi-structured interviews with them, using a topic guide developed by the study team (AMS, CH, FG and Cynthia Ochieng), as informed by the literature and discussions with the study's clinical co-investigators. Analysing these initial data, we recognised that ReSPECT conversations were not confined to clinical encounters during ward

	rounds. Additionally, based on these preliminary results, AMS, CH, FG and KE refined the existing topic guide and developed new questions. We enclose the interview guide as a supplementary file.
Comment 4	While a fair amount of space was spent in the methods on these interviews, it is unclear how they were used in the final analysis of this paper. The methods section states “This paper focuses on ReSPECT conversations that did not occur” and then goes on to state that “She triangulated data drawn from her fieldnotes with data drawn from her formal interviews with the clinicians involved in these cases, in which they reflected on why these conversations did not occur.” Does that mean that only clinician interviews that were involved with the cases were included (as is suggested by the methods section)? Or were all clinician interviews included in the triangulation, picking out those relevant quotes from all interviews
Response 4	Only clinician interviews that were involved with the selected cases were used in the manuscript; we now clarify this in the manuscript (p. 5): “She triangulated data drawn from her fieldnotes with data drawn from her formal interviews with the clinicians involved in these six cases, in which they reflected on why these conversations did not occur.”
Comment 5	More information about the six cases described is needed: Were they all from the same site? From the same specialty/discipline? What level of training were the clinicians involved? Much of this information can be gleaned from the narrative descriptions in the results section but it is inconsistently reported. A table summarizing this information would answer many of these questions and allow for a better overall understanding of the context of the six case studies. Alternatively, a short summary paragraph at the beginning of the results section may also suffice for relaying this information.
Response 5	We have added a table that summarizes the key characteristics of the clinicians involved in the six cases (Table 2).
Comment 6	Besides the statement “She triangulated data drawn from her fieldnotes with data drawn from her formal interviews with the clinicians involved in these cases, in which they reflected on why these conversations did not occur,” there was little information on the analysis of the data. Were these cases discussed at meetings (such as at a coding meeting)? If so, please describe the make-up of this group. Was there any software used to assist in the coding of data (ie: Atlas, Dedoose)? Were there any other techniques to enhance trustworthiness and credibility of data analysis such as member checking? Or was the coding, triangulation and ultimate development of the “thick description” done purely by one person (which is what the methods section suggests)?
Response 6	We now clarify this in the manuscript as follows (p. 5): “While data analysis was performed by KE, in keeping with ethnographic analysis methods, the findings were discussed and refined in team discussions which included all co-authors. Data analysis software was not used.”
Comment 7	Thank you for the opportunity to complete this review and I look forward to seeing this article in print.
Response 7	Thank you for your helpful comments; we hope you will find our manuscript has improved as a result of the revision process.

Reviewer: 2
Dr. Ting Hway Wong, Singapore General Hospital

Comment 1	Thank you for this beautifully conducted study (both the submitted draft and the larger overall study). As an emergency surgeon with a specific interest in end of life issues, this is an important issue to address.
Response 1	Thank you for your encouraging feedback on our work.
Comment 2	My main concern with the study as it stands is that it only shows the barriers to ECTP (what some of us surgeons would call a "negative demonstration"), which is meaningless without counterbalance with examples of when they were successful. If, for example, another surgeon had used exactly the same reasons to push for ECTP (eg "patient is in a bad way and I worry we don't have much time to have this conversation together" / "we are expecting to be called to another major emergency and hope we could catch some time for this conversation before our schedule is full"), then the ethnographic findings would only paint half the picture. I understand word count may have been a problem, but a workaround should be found (e.g. part 1 Why they do not discuss ECTP and part 2 Why they do).
Response 2	Thank you for this opportunity to explain our approach to this manuscript. In our earlier work, we have analysed clinicians' experiences of conducting ECTP conversations (Eli et al., 2020) as well as when, why, and how clinicians initiate these conversations (Eli et al., 2021). We have therefore addressed your question elsewhere, and have now added references to these papers, to ensure that readers can place the findings we report in the current manuscript into context. However, in this manuscript, our focus is specifically on instances where conversations were delayed or avoided. We believe this is an important focus, given the centrality of 'non-conversations' in the study findings, and considering the tendency in academic research to publish only positive results. We hope this clarifies our approach. We've added an explanation to the methods, as follows (p. 5): "This paper focuses on ReSPECT conversations that did not occur. We have previously described the range of ReSPECT conversations that occur following acute hospital admissions, and the factors that facilitate these conversations (Eli et al., 2021). However, because waiting times dominated the collection of data, and because waiting for a planned conversation did not always lead to an actual conversation, focusing on ReSPECT conversations alone as the key object of research would miss the realities between conversation events."

VERSION 2 – REVIEW

REVIEWER	Kathleen O'Neill Yale-New Haven Hospital
REVIEW RETURNED	23-Mar-2021
GENERAL COMMENTS	The authors have adequately addressed all major concerns from the review. I look forward to seeing the article in print.